# Large Language Model Guided Dynamic Branching Rule Scheduling in Branch-and-Bound

## Abstract

Branch-and-bound (B&B) is a core technique in state-of-the-art mixed integer linear program (MILP) solvers. It reformulates an MILP into a systematic tree search and recursively partitions it into subproblems using various hard-coded heuristics, among which the branching rule plays a central role. Different branching rules yield distinct search trajectories and performance outcomes, making their selection a decisive factor in solver performance. Traditionally, the configuration of the branching rule heavily relies on expert knowledge: a rule is manually configured for a given problem and applied throughout the entire B&B process, or predefined to switch at certain depths. Such approaches fail to adapt to the evolving structure of the search tree, which often leads to suboptimal branching decisions and inefficient exploration of the search space. More recently, learning-based branching policies have been proposed to automate branching decisions using feature representations, but they often involve costly training pipelines and exhibit poor generalization across heterogeneous problem types. In this work, we propose a large language models (LLMs)-guided approach to dynamically schedule the branching rule throughout the B&B process. The term *dynamic scheduling* refers to (i) identifying the problem type and scale at the initial stage to select an appropriate starting rule, and (ii) monitoring the evolving state of the search tree during solving to adaptively decide when and which branching rule to switch. By leveraging the extensive prior knowledge embedded in LLMs, our method eliminates dependence on human-crafted heuristics, removes the need for dedicated training, and achieves zero-shot generalization across diverse problem types. Experiments on benchmark instances demonstrate that our method shows great potential and achieves competitive performance with state-of-the-art baselines in terms of solving efficiency.

## 1 Introduction

Mixed integer linear program (MILP) is a powerful method for modeling combinatorial optimization problems and have been widely applied in real-world domains (Kacem et al., 2025; Wolf, 2011). The predominant method for solving MILPs to global optimality is the Branch-and-bound (B&B) algorithm, which adopts a divide-and-conquer strategy. Specifically, B&B reformulates an MILP into a systematic tree search and recursively partitions it into subproblems using the branching rule. A variety of branching rules, such as most infeasible branching (Mostinf), pseudocost branching (Pscost), and fullstrong branching (Fullstrong) have been implemented in modern solver. These rules differ in the way they select branching variables and prioritize subproblems for exploration, resulting in distinct search trajectories and solver performance. Each rule is suited to different scenarios—for example, strong branching is effective for small-scale problems or at the root, whereas pseudocost branching is better for large-scale problems once sufficient warm-up has been achieved for stable performance.

**Motivation** For a given MILP problem, the choice of an appropriate branching rule has a decisive impact on practical solving performance (Achterberg & Wunderling, 2013). As shown in Figure 1(a), for a set covering problem with five different branching rules, each executed five times with different random seeds, the solving times (normalized) vary substantially across rules, highlighting the sensitivity of solver performance to rule selection. In practice, rule selection largely relies on expert knowledge of the problem structure. Once selected, the initial branching rule is applied throughout the entire B&B process. However, as optimization progresses, the structure of the search tree evolves, and adhering to a single rule is often suboptimal for the solving process. Figure 1(b) shows that switching rules during B&B produces a superadditive effect, reducing branching steps more effectively than any individual rule alone. Whereas, both the timing and the

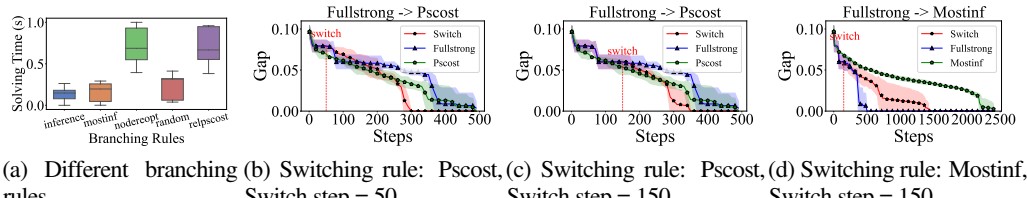

(a) Different branching rules

(b) Switching rule: Pscost, Switch step = 50

(c) Switching rule: Pscost, Switch step = 150

(d) Switching rule: Mostinf, Switch step = 150

Figure 1: Preliminary experiment results on capacitated facility location benchmark.

choice of rule have a significant impact on solving performance. For example, switching to the Pseudocost rule at step 50 reduces the number of branching steps more effectively than switching at step 150, as shown in Figures 1(b) and (c). Figures 1(c) and (d) show that switching to the Pseudocost rule results in faster convergence than switching to the Mostinf rule. Further details on the impact of different branching rules on MILP problems, as well as their effects on the B&B tree search process, are provided in Appendix A.

Based on the above preliminary results, we observe that rule selection/switching play a critical role in solving performance. However, effective rule selection and switching rely on interpreting search-tree dynamics and understanding how different branching rules behave under those dynamics, a process that typically requires considerable expert expertise. Therefore, we aim to address the following challenges in order to minimize reliance on expert knowledge when switching branching rules during the B&B procedure: (i) selecting an appropriate branching rule for a given problem and (ii) determining when to switch and which rule to adopt. These two challenges are critical yet underexplored questions. In this work, we propose dynamic branching rule scheduling powered by LLMs throughout the B&B procedure. At the initial stage, we leverage the broad knowledge of LLMs to select a suitable branching rule according to the problem type and scale. During solving, the evolving B&B search tree is translated into a linguistic representation, enabling LLMs to assess the state of the tree and provide guidance on whether to switch branching rules for more effective solving. To ensure efficiency and mitigate the overhead of LLM queries, we adopt an asynchronous real-time interaction with LLMs, combined with a multi-LLM voting mechanism for branching rule selection. We evaluate our approach on four classes of NP-hard MILP problems: set covering, combinatorial auctions, capacitated facility location, and maximum independent set. We compare it against previously proposed state-of-the-art methods and implement our approach in SCIP (Bestuzheva et al., 2023), a modern open-source solver. The results demonstrate competitive performance relative to existing models and highlight the potential of our research for building more intelligent solvers. In summary, our contributions are as follows:

- We investigate the important yet underexplored problem of dynamic branching rule scheduling, which has traditionally relied heavily on expert knowledge, and propose a novel scheduler that dynamically selects appropriate rules during the B&B search process.

- We design effective prompting strategies that (i) guide branching rule selection at the initial stage based on problem type and scale, and (ii) adaptively guide rule switching during solving by leveraging the evolving dynamics of the search tree.

- We conduct extensive experiments to validate the effectiveness of the proposed scheduler and demonstrate the potential of LLMs in supporting MILP solving.

## 2 RELATED WORK

### 2.1 TRADITIONAL BRANCHING RULES

Branching is a core step in branch-and-bound for mixed-integer programming (MIP), where selecting the fractional variable at each node critically affects search efficiency. Traditional strategies follow a generic scheme: evaluate objective degradation in both child LP relaxations, assign a score, and branch on the best candidate. This scheme underlies many classic rules, such as most-infeasible branching (Achterberg et al., 2005), which selects the variable closest to 0.5 but rarely outperforms random branching. Over time, more sophisticated scoring functions were introduced that estimate the degradation in the LP relaxation objective when branching on a candidate variable. These rules can generally be expressed in a unified framework that balances the minimum and maximum predicted objective changes of the child nodes, highlighting their conceptual connections and limitations. Building on this foundation, several influential strategies have been

developed. Pseudocost branching (Linderoth & Savelsbergh, 1999) estimates the effect of future branchings based on accumulated historical outcomes, offering efficiency once sufficient data is gathered but lacking reliability near the root. Strong branching (Applegate et al., 1995) explicitly tests candidate variables by temporarily solving child LPs, producing high-quality choices at the cost of significant computational overhead. To combine their advantages, hybrid rules (Lin & Schrage, 2009) apply strong branching at shallow depths before switching to pseudocosts, while pseudocost with strong initialization uses strong branching selectively to bootstrap missing statistics. The most effective generalization, reliability branching (Achterberg, 2007), adaptively decides when pseudocosts are "reliable" enough, invoking strong branching only when necessary. Extensive computational studies demonstrate that reliability branching consistently outperforms depth-based hybrids, achieving performance close to full strong branching without incurring its high costs.

## 2.2 MACHINE LEARNING-BASED BRANCHING RULES

In recent years, machine learning techniques have been actively explored to automate the search process in branch-and-bound (Huang et al., 2022; Mattick & Mutschler, 2024; Achterberg et al., 2005; Tang et al., 2020). In ML-based B&B, the core idea is to learn effective policies that guide the search more efficiently, which can be broadly categorized into learning to select variables and learning to select nodes. The former focuses on identifying the most promising branching variable at each step, while the latter emphasizes selecting the next node to explore in the search tree.

For learning to select variables, an instance-specific learning-to-rank approach (Khalil et al., 2016) imitates strong branching decisions, while regression models trained on strong branching scores across families of similar instances (Alvarez et al., 2017) provide variable selection guidance. A nearly optimal mixture of branching rules (Balcan et al., 2018) emerges by learning under distributional assumptions over problem instances. A bipartite graph formulation of MILPs combined with imitation learning (Gasse et al., 2019) enables graph convolutional neural network (GCN) to handle variable selection in arbitrarily sized problems. Reinforcement learning is also applied to B&B by modeling it as a tree Markov decision process (tMDP) (Scavuzzo et al., 2022). A hybrid model (Gupta et al., 2020) replaces costly graph networks with multi-layer perceptrons (MLPs) except at the root node, making the approach suitable for CPU-limited environments. A joint use of neural diving and neural branching (Nair et al., 2020) achieves significant improvements in both runtime and the average primal–dual gap. Although these methods demonstrate strong performance, their efficiency gains are typically confined to specific classes of MILPs. To improve generalization across heterogeneous problems, branching policy that parameterize the state of branch-and-bound search trees and imitate SCIP's default relpscost rule (Zarpellon et al., 2021) serves as effective expert policies due to their tree-oriented focus. Extending this idea, tree-based representations combined with transformer-based policy networks (Lin et al., 2022) provide more expressive feature extractors while still imitating the relpscost rule.

Learning-based node selection plays an important role in branch-and-bound. (He et al., 2014) designs adaptive node search orders with imitation learning that generalize across different classes of problems solved by branch-and-bound. (Song et al., 2018) leverages improved traces constructed from its own roll-outs to iteratively scale policies to larger problem instances. (Khalil et al., 2022) exploits variable–constraint bipartite graphs to predict variable biases and guide solver decisions such as node selection, outperforming heuristic baselines. Tree-level reinforcement learning with graph neural networks enables node selection policies that consider entire tree states rather than isolated nodes, improving efficiency and generalization (Mattick & Mutschler, 2024). (Zhang et al., 2025) introduces a tripartite graph representation combined with reinforcement learning to capture sufficient information from the branch-and-bound tree and evaluate node quality more comprehensively.

## 2.3 DISCUSSION

We briefly compare the aforementioned methods across four dimensions in Table 1. Traditional branching rules encode hard-coded expert heuristics that, while generally applicable across problems, lack the ability to sense the search tree state and schedule different rules. ML-based branching leverages large collections of branching examples to train intelligent policies, yet collecting and curating such datasets is time-consuming, and policies trained on one problem type often fail to generalize to unseen types. These limitations motivate our LLM-based branching rule scheduler, which uses problem descriptors (e.g., type and scale) for zero-shot initialization and employs tree state-aware prompts (including depth, gap trends, cutoff ratios, and candidate entropy) to recommend when and to which rule to switch. Crucially,

Table 1: Brief comparison of the different branching rules.

| | Model | Gen.[1] | Data Inden.[2] | Rule Sched.[3] | Tree-aware.[4] |
|---|---|---|---|---|---|
| Traditional | Fullstrong | ✓ | ✓ | ✗ | ✗ |
| | Mostinf | ✓ | ✓ | ✗ | ✗ |
| | Pseudocost | ✓ | ✓ | ✗ | ✗ |
| ML-based | (Khalil et al., 2016) | ✗ | ✗ | ✗ | ✗ |
| | (Gasse et al., 2019) | ✗ | ✗ | ✗ | ✗ |
| | (Gupta et al., 2020) | ✗ | ✗ | ✗ | ✗ |
| | (Zarpellon et al., 2021) | ✓ | ✗ | ✗ | ✗ |
| | (Lin et al., 2022) | ✓ | ✗ | ✗ | ✓ |
| | (Zhang et al., 2025) | ✗ | ✗ | ✗ | ✓ |
| LLM-based | **Ours** | ✓ | ✓ | ✓ | ✓ |

[1] **Generalization**: Ability to handle unseen problem types and maintain stable performance.

[2] **Data Independence:** Performance does not rely on the feature engineering or richness of training data.

[3] **Rule Scheduling:** Ability to dynamically adjust branching rules during the search process.

[4] **Tree-awareness:** Ability to incorporate the evolving state of the search tree into decision-making.

this approach requires no training pipeline and generalizes across unseen problem types by leveraging the broad prior knowledge embedded in large language models.

## 3 PRELIMINARIES

### 3.1 BRANCH-AND-BOUND

A mixed-integer linear programming (MILP) instance can be written as:

$$\min \boldsymbol{c}^\top \boldsymbol{x} \quad \text{s.t.} \quad \boldsymbol{A}\boldsymbol{x} \leq \boldsymbol{b}, \boldsymbol{l} \leq \boldsymbol{x} \leq \boldsymbol{u}, x_j \in \mathbb{Z}, \forall j \in \mathcal{I}, \tag{1}$$

where $\boldsymbol{A} \in \mathbb{R}^{m \times n}$ is the constraint matrix, $\boldsymbol{c} \in \mathbb{R}^n$ is the cost vector, $\boldsymbol{b} \in \mathbb{R}^m$ is the right-hand-side constraint values, $\boldsymbol{l}, \boldsymbol{u}$ denote the variable bounds, and $\mathcal{I}$ is the index set of integer-constrained variables. The branch-and-bound (B&B) algorithm solves such problems by iteratively exploring subsets of the feasible region. At each node, a linear programming (LP) relaxation is solved by ignoring integrality constraints, producing a solution $\boldsymbol{x}^*$ that yields a lower bound on the MILP objective. If $\boldsymbol{x}^*$ happens to satisfy all integrality constraints, it is a feasible MILP solution and its objective value becomes an upper bound. Otherwise, a fractional variable $x_j^*$ $(j \in \mathcal{I})$ is selected for branching, and the problem is split into two subproblems with additional constraints $x_j \leq \lfloor x_j^* \rfloor$ and $x_j \geq \lceil x_j^* \rceil$, where $\lfloor \cdot \rfloor$ and $\lceil \cdot \rceil$ denote the floor and ceiling operators. During the search, any node whose lower bound exceeds the current best upper bound is discarded, a process called pruning. This branching–bounding–pruning cycle continues until all nodes are either solved or pruned, at which point the best feasible solution found is guaranteed to be globally optimal (or within a specified optimality tolerance).

### 3.2 BRANCHING RULE

In branch-and-bound (B&B), branching involves two decisions: node selection, which determines the next node to explore, and variable selection, which chooses the fractional variable to branch on at a given node. While both are crucial, our work focuses on variable selection, where branching rules guide the evaluation and ranking of candidate variables. Traditional rules (Achterberg et al., 2005; Linderoth & Savelsbergh, 1999; Applegate et al., 1995) rely on hard-coded heuristics, which are either inefficient or effective only in limited scenarios. To strike a better balance, recent neural network–based approaches learn branching policies by collecting large numbers of branching examples for training. However, these methods often

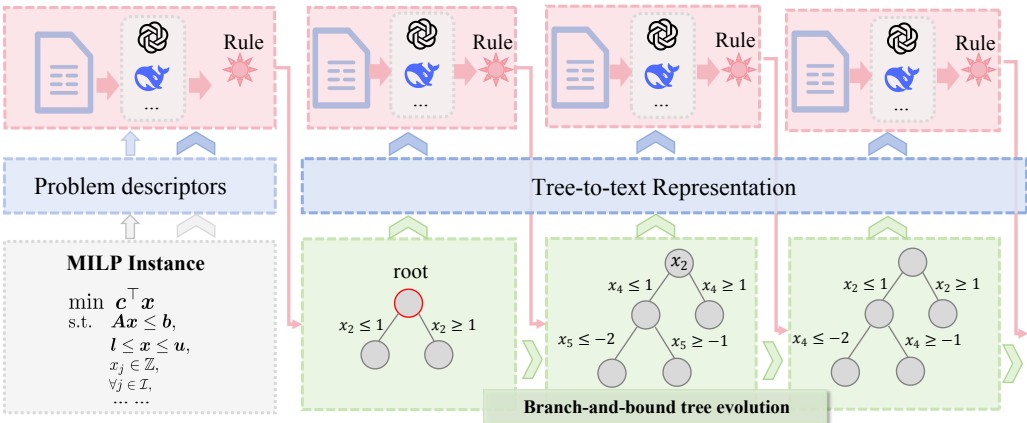

Figure 2: The overview of our approach.

fail to generalize to unseen problem types, and collecting training data is time-consuming. Orthogonal to these approaches, our goal is not to design or train yet another branching rule from scratch, but to study how to schedule existing rules. Since different branching rules are suited to different search tree conditions, our objective becomes identifying the state of the search tree and knowing which rules are most effective under which circumstances, so as to schedule appropriate rules at different stages of the search.

## 4 METHODOLOGY

We propose a dynamic branching rule scheduling approach powered by large language models to enhance the B&B algorithm, as illustrated in Figure 2. First, during initialization, LLMs recommend an appropriate branching rule based on the problem type and scale (Section 4.1). Second, throughout the solving process, the evolving search tree is continuously monitored and represented in a tree-to-text linguistic form, enabling LLMs to decide whether to switch rules and which rule to adopt (Section 4.2). Finally, to ensure both efficiency and robustness, the framework incorporates an asynchronous multi-LLM decision mechanism, where multiple LLMs monitor the tree state in parallel, propose decisions, and consolidate them into the final choice through a voting scheme (Section 4.3).

### 4.1 INITIAL BRANCHING RULE

Selecting an appropriate initial branching rule is crucial for guiding the subsequent tree search in B&B. The performance of different rules varies significantly depending on the problem type (e.g., set covering, maximum independent set) and scale (e.g., number of variables and constraints). Traditionally, specifying the branching rule relies heavily on expert knowledge and manual configuration. To reduce this dependency and improve solver intelligence, we leverage the broad prior knowledge of LLMs to replace human expertise in the initialization stage. Specifically, we extract heuristic guidelines from solver documentation and prior literature. Let $\mathcal{R} = \{r_1,...,r_M\}$ denote the set of candidate branching rules, where each rule $r_i$ is associated with a heuristic guideline function. Given an MILP instance $\mathcal{P}$, we extract its high-level descriptors, such as problem type $t$ (e.g., set covering, facility location) and problem scale $s \in \mathbb{R}^+$ (e.g., number of variables and constraints). The former can be optionally specified by the user when providing the instance, while the latter can be obtained directly through simple statistical analysis. We then formulate prompts and query a set of LLMs:

$$\Pi = \{\pi_1,...,\pi_M\}, \quad \pi_i(\texttt{Prompt}(t,s,\mathcal{R})) \mapsto r_i, \qquad (2)$$

where $\Pi$ denotes a collection of $M$ LLMs, $\pi_i(\cdot)$ represents the $i$-th LLM in the ensemble, and $r_i \in \mathcal{R}$ is the branching rule it selects from the candidate set $\mathcal{R} = \{r_1,...,r_M\}$. Here, $\texttt{Prompt}(\cdot)$ refers to the constructed input prompt based on the problem descriptors, with the full template provided in Appendix B. We extract prior knowledge of each rule from solver documentation and the existing literature, incorporating it as prompt augmentation to guide LLM decision-making. The final branching rule is then determined through majority voting across all LLMs:

$$r^* = \text{Vote}\big(\{r_1,...,r_M\}\big), \qquad (3)$$

where $\text{Vote}(\cdot)$ selects the rule with the highest frequency among the LLM outputs. This ensemble mechanism mitigates the impact of individual LLM hallucinations and enhances the robustness of the initial rule recommendation.

## 4.2 DYNAMIC RULE SWITCHING

The structure of the B&B search tree evolves dynamically as solving progresses, and a fixed branching rule applied throughout the process cannot achieve optimal performance at all stages. To address this limitation, we introduce a mechanism that adaptively switches branching rules based on the evolving dynamics of the search tree. Specifically, we convert subtrees of the search tree into linguistic representations and leverage LLMs to evaluate whether the current rule remains suitable or should be switched to a more effective alternative given the observed metrics.

**Tree-to-text representation** At a given branching step $k$, we construct a decision subtree $\mathcal{T}_k^L$ consisting of the current node and its $L$ most recent ancestors. We maintain a queue of length $L$ that records the explored nodes and their corresponding metrics $\mathcal{T}_k^L = \{v_{k-L+1}, v_{k-L+2}, ..., v_k\}$ during the B&B search, as illustrated in Step 1 of Figure 3, where $v$ denotes the explored nodes and their corresponding metrics. Whenever a re-evaluation is triggered, the node information in the queue is retrieved and converted into a textual representation, as shown in Step 2 of Figure 3. Specifically, we store six key features that characterize the search state: *node depth, number of explored nodes, relative optimality gap, cutoff ratio, domain reduction strength, number of branching candidates, and candidate entropy*. These features provide a comprehensive description of whether the current branching rule remains effective for guiding the search.

**Switching instruction** We denote the textual form of the tree state as $\text{text}(\mathcal{T})$, which is then converted into a prompt for the LLM. In addition, the prompt incorporates decision rubrics that evaluate rule effectiveness through sliding-window comparisons of the stored metrics. Specifically, the LLM is instructed to reason about the step-wise changes $\Delta(v[j])$, where $j$ indexes the six recorded metrics, and to infer from their trends whether the current branching rule continues to enable effective search. If stagnation or inefficiency is detected, the LLM is guided to decide whether the rule should be switched. When a switch is required, the LLM selects a new branching rule from the available set by jointly considering the recent search tree dynamics (e.g., depth, number of candidates, entropy trends) and the heuristic guidelines associated with each rule, and then activates the chosen rule for subsequent exploration.

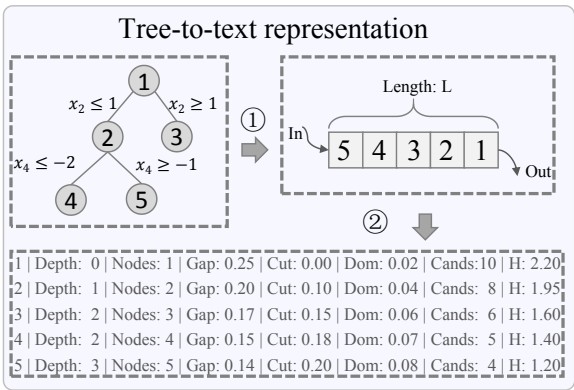

Figure 3: Illustration of tree-to-text representation

During solving, we likewise employ an ensemble of LLMs with a voting scheme to determine branching rule switches. The interaction with the LLMs is conducted asynchronously, which prevents solver delays caused by waiting for model responses.

## 4.3 ASYNCHRONOUS MULTI-LLM DECISION MECHANISM

Although LLMs can provide valuable guidance for branching rule selection and switching, their inference introduces latency and potential hallucination (Ho et al., 2025), both of which may degrade solver performance. To address these challenges, we design an asynchronous multi-LLM decision mechanism that improves efficiency while enhancing robustness.

**Asynchronous monitoring** Instead of blocking the solver while waiting for LLM responses, we launch independent asynchronous LLM processes that continuously monitor the evolving search tree. At each interaction, the solver sends the encoded text subtree $\mathcal{T}$ to the LLM ensemble $\Pi$. The LLMs reason in parallel about whether a rule switch is needed, while the solver continues its search using the last validated rule. Once the new recommendations arrive, the scheduler updates the rule selection without interrupting

the ongoing solving process. Although asynchronous responses may arrive a few steps later than the current tree state, this does not invalidate the feedback: given the large number of steps in the solving process, a slight delay still provides guidance that remains relevant to the evolving search tree.

**Voting scheme** To mitigate the impact of hallucination and reduce bias from individual models, we aggregate the outputs of multiple LLMs through majority voting. Let $\{r_1, r_2, ..., r_M\}$ be the set of candidate rules from the LLM ensemble. The final decision rule is formulated as:

$$r^* = \text{Vote}\big(\{r_1, ..., r_M\}\big) = \arg\max_{r \in \mathcal{R}} \sum_{i=1}^{M} \mathbb{I}[r_i = r], \tag{4}$$

where $\mathbb{I}[\cdot]$ is the indicator function. This ensures that only rules supported by the majority are adopted, thereby reducing the influence of inconsistent or erroneous outputs from individual LLMs. By combining asynchronous interaction and ensemble voting, the proposed mechanism simultaneously minimizes solver slowdown due to LLM inference and increases decision robustness against hallucinations. This design provides a practical and reliable way to integrate LLM guidance into the B&B process.

## 5 EXPERIMENTS

We conduct comparative experiments against three representative machine learning baselines and SCIP's default branching strategy to evaluate the effectiveness of our method, and further perform an ablation study to examine the impact of our design choices.

### 5.1 SETUP

Our experiments are conducted on a machine equipped with an Intel(R) Xeon(R) Silver 4214R CPU @ 2.40GHz and 256GB DDR4 memory, with GPUs disabled. The LLM ensemble consists of qwq2-32b, Claude-3.5, Gemini-2.5, DeepSeek v3.1, and Qwen3-30b-Thinking. All experiments are performed using SCIP 6.0.1 with a one-hour time limit, while all other parameters are kept at their default values following (Gasse et al., 2019; Scavuzzo et al., 2022) to ensure fairness and reproducibility.

#### 5.1.1 BENCHMARKS

We evaluate our approach on four NP-hard benchmarks widely used in integer programming. The first is the set covering problem (Balas & Ho, 2009), with evaluation conducted on instances of size 500 (Easy), 1,000 (Medium), and 2,000 (Hard). The second is the combinatorial auction problem (Leyton-Brown et al., 2000), with 100 items/500 bids (Easy), 200/1,000 bids (Medium), and 300/1,500 bids (Hard). The third is the capacitated facility location problem (Cornuéjols et al., 1991), evaluated on 100 (Easy), 200 (Medium), and 400 (Hard) customers. The final benchmark is the maximum independent set problem (Bergman et al., 2016), with evaluation on graphs of 500 (Easy), 1,000 (Medium), and 1,500 (Hard) nodes. These benchmarks are both challenging for state-of-the-art solvers and representative of practical integer programming tasks.

#### 5.1.2 BASELINES

We benchmark our method against SCIP's default state-of-the-art branching rule, reliability pseudocost (RPB), a variant of hybrid branching (Achterberg & Berthold, 2009). In addition, we compare with four learning-based branchers: SVMRANK, a learning-to-rank approach from (Khalil et al., 2016) built on SVMrank (Joachims, 2002), and LMART, a LambdaMART-based variant proposed in (Hansknecht et al., 2018). Both SVMRANK and LMART rely on the original feature set introduced in (Khalil et al., 2016). We further include two policy-learning approaches: GCN (Gasse et al., 2019), which applies imitation learning with graph convolutional networks (GCN), and tMDP (Scavuzzo et al., 2022), which leverages reinforcement learning within a temporal Markov decision process framework. Although the GCN and tMDP methods support GPU acceleration, all other models in our comparison cannot be accelerated on GPUs. To ensure fairness, we therefore restrict both GCN and tMDP to CPU execution. Moreover, in reporting the results, we ignore the time required to train the GCN and tMDP branching policies, and only account for the solving time after training when the learned policies are applied during the branch-and-bound process.

### 5.1.3 EVALUATION METRICS

Evaluation is conducted across three difficulty levels (Easy, Medium, Hard). For the Easy and Medium settings, we use 100 instances each, while for the Hard setting we only evaluate on 20 instances due to the higher computational cost. We adopt standard MILP benchmarking metrics: (i) the mean solving time in seconds (Time), including unsolved cases; (ii) the proportion of instances solved to optimality (Optimal) within the time limit (e.g., "90/100" means 90 out of 100 instances were solved optimally, while 10 were not); and (iii) the number of wins, i.e., how often a method achieves the fastest runtime among the compared methods (Wins). For clarity, we also report the average per-instance standard deviation. For example, "14.37 ± 1.5% nodes" indicates an average solving time of 14.37 seconds with a mean variability of 1.5% across the solved instances.

Table 2: Methods evaluation on separate instances in terms of solving time, number of optimal instances and number of wins (fastest method)

| Model | Easy | | | Medium | | | Hard | | |
|---|---|---|---|---|---|---|---|---|---|
| | Time ↓ | Optimal ↑ | Wins ↑ | Time ↓ | Optimal ↑ | Wins ↑ | Time ↓ | Optimal ↑ | Wins ↑ |
| | | | | Set Covering | | | | | |
| RPB | 14.37 ± 1.5 % | 100/100 | 1/100 | **141.98** ± 11.3% | 100/100 | **46**/100 | 2479.20 ± 7.9% | 16/20 | 7/20 |
| SVMRANK | 11.15 ± 2.4% | 100/100 | 0/100 | 175.47 ± 13.1% | 100/100 | 0/100 | 3008.28 ± 7.7% | 14/20 | 0/20 |
| LMART | **10.06** ± 3.7% | 100/100 | 14/100 | 147.77 ± 9.8% | 100/100 | 9/100 | 3037.52 ± 8.6% | 14/20 | 1/20 |
| GCN | 13.26 ± 3.1% | 100/100 | 0/100 | 248.77 ± 9.7% | 100/100 | 0/100 | 3334.24 ± 5.3% | 17/20 | 0/20 |
| tMDP | 15.17 ± 3.3% | 100/100 | 0/100 | 256.51 ± 8.3% | 100/100 | 0/100 | 3617.32 ± 6.5% | 16/20 | 0/20 |
| Ours | 10.47 ± 1.7% | 100/100 | **84**/100 | 181.32 ± 3.98% | 100/100 | 45/100 | **2394.98** ± 8.1% | **18**/20 | **12**/20 |
| | | | | Combinatorial Auction | | | | | |
| RPB | 3.82 ± 2.3% | 100/100 | 7/100 | 28.12 ± 7.5% | 100/100 | **52**/100 | **108.68** ± 8.1% | 20/20 | **12**/20 |
| SVMRANK | 3.40 ± 1.7% | 100/100 | 0/100 | 39.49 ± 11.6% | 100/100 | 0/100 | 569.50 ± 9.0% | 19/20 | 0/20 |
| LMART | 2.57 ± 1.4% | 100/100 | 18/100 | 25.24 ± 5.5% | 100/100 | 17/100 | 443.65 ± 3.1% | 20/20 | 0/20 |
| GCN | 3.78 ± 2.8% | 100/100 | 0/100 | 46.96 ± 10.2% | 100/100 | 0/100 | 854.63 ± 7.7% | 19/20 | 0/20 |
| tMDP | 4.11 ± 2.6% | 100/100 | 0/100 | 53.37 ± 7.2% | 100/100 | 0/100 | 717.90 ± 7.1% | 19/20 | 0/20 |
| Ours | **2.18** ± 1.6% | 100/100 | **75**/100 | 35.24 ± 5.9% | 100/100 | 31/100 | 116.51 ± 4.4% | 20/20 | 8/20 |
| | | | | Capacitated Facility Location | | | | | |
| RPB | 72.70 ± 10.7% | 100/100 | 24/100 | 291.88 ± 6.4% | 100/100 | 25/100 | 1099.33 ± 4.1% | 20/20 | 8/20 |
| SVMRANK | 139.06 ± 13.2% | 100/100 | 0/100 | 366.63 ± 14.3% | 100/100 | 7/100 | 1104.27 ± 6.9% | 19/20 | 0/20 |
| LMART | 169.37 ± 11.6% | 100/100 | 0/100 | 378.33 ± 11.0% | 100/100 | 10/100 | 1170.29 ± 4.0% | 20/20 | 0/20 |
| GCN | 223.92 ± 9.6% | 100/100 | 0/100 | 541.55 ± 11.0% | 100/100 | 0/100 | 1390.45 ± 9.6% | 18/20 | 0/20 |
| tMDP | 251.65 ± 18.9% | 100/100 | 0/100 | 541.55 ± 14.9% | 100/100 | 0/100 | 1455.07 ± 5.9% | 19/20 | 0/20 |
| Ours | **57.05** ± 12.8% | 100/100 | 76/100 | **283.47** ± 5.8% | 100/100 | 58/100 | 1079.44 ± 1.2% | 20/20 | **12**/20 |
| | | | | Maximum Independent Set | | | | | |
| RPB | 12.87 ± 4.8% | 100/100 | 24/100 | **299.00** ± 7.6% | 100/100 | 62/100 | 2806.84 ± 4.9% | 10/20 | 8/20 |
| SVMRANK | 16.28 ± 8.7% | 100/100 | 3/100 | 773.37 ± 11.8% | 100/100 | 4/100 | 2866.56 ± 9.4% | 14/20 | 1/20 |
| LMART | **9.94** ± 4.0% | 100/100 | 12/100 | 715.91 ± 8.8% | 100/100 | 10/100 | 2922.34 ± 6.7% | 14/20 | 1/20 |
| GCN | 22.09 ± 5.6% | 100/100 | 1/100 | 2028.37 ± 9.4% | 100/100 | 0/100 | 3600.03 ± 7.2% | 0/20 | 0/20 |
| tMDP | 12.86 ± 12.1% | 100/100 | 0/100 | 1989.29 ± 10.1% | 100/100 | 0/100 | 3521.22 ± 7.2% | 1/20 | 0/20 |
| Ours | 12.41 ± 3.7% | 100/100 | **60**/100 | 430.59 ± 9.2% | 100/100 | 24/100 | **2742.75** ± 8.1% | 10/20 | **10**/20 |

### 5.1.4 RESULTS

**Main rusults** The comparison results are summarized in Table 2. On the easy and medium instances, all models are able to reach the optimal solution (100/100 success rate). In terms of the time metric, we observe that on easy instances both LMART and our proposed method perform competitively. LMART achieves good results because its tree-based ranking model is particularly effective when the problem size is relatively small and the feature distributions are well captured. Our method performs particularly well at the Hard level, since the search tree at this scale requires exploring a much larger space. By dynamically switching branching rules during the search process, our approach is able to adapt more effectively, leading to superior performance. Notably, since optimal solutions can be reached with only a few branching steps in easy level, our approach does not perform dynamic rule switching and instead relies solely on the initial rule selection, highlighting the competitiveness of our initialization selection. We further observe that GCN and tMDP, the most advanced learning-based approaches, lose much of their advantage once GPU acceleration is removed. Their efficiency drops substantially and even deteriorates on certain datasets—for instance, on the medium-level facility location benchmark, both methods take an order of magnitude longer solving time, and on the hard level they almost fail to reach optimal solutions at all. This performance gap is especially notable given that our reported results exclude the training time required for these neural models. These findings indicate that the effectiveness of current learning-based methods depends heavily on GPU acceleration, limiting their practical applicability in resource-constrained environments.

Table 3: Ablation study results.

| Model | Easy | | | Medium | | | Hard | | |
|---|---|---|---|---|---|---|---|---|---|
| | Time ↓ | Optimal ↑ | Wins ↑ | Time ↓ | Optimal ↑ | Wins ↑ | Time ↓ | Optimal ↑ | Wins ↑ |
| | | | | Capacitated Facility Location | | | | | |
| RPB | 70.70 ± 10.7% | 100/100 | 10/100 | 291.88 ± 6.4% | 100/100 | 27/100 | 1099.33 ± 4.1% | 20/20 | 7/20 |
| Stage1 | 76.28 ± 8.7% | 100/100 | 5/100 | 307.56 ± 5.8% | 100/100 | 2/100 | 1135.25 ± 2.5% | 20/20 | 0/20 |
| Stage2 | 62.94± 4.0% | 100/100 | 10/100 | 285.19 ± 6.4% | 100/100 | 20/100 | 1082.33 ± 8.6% | 20/20 | 3/20 |
| Ours | **57.05** ± 12.8% | 100/100 | **75**/100 | **283.47** ± 5.8% | 100/100 | **51**/100 | **1079.44** ± 1.2% | 20/20 | **10**/20 |

**Abalation study** To validate the effectiveness of the two stages in our proposed framework, we conduct an ablation study by isolating each component as Table 3 shown. Specifically, we retain only the initial rule recommendation while removing dynamic rule switching, denoted as Stage1, and we fix the initial rule to fullstrong while enabling switching during solving, denoted as Stage2. We observe that at the Easy level, Stage1 still outperforms the solver's default RPB branching rule. However, at the Medium and Hard levels, Stage1 performs poorly, whereas Stage2 demonstrates strong results. This is because the search space at Medium and Hard levels is substantially larger, and relying solely on the initial choice of branching rule is insufficient to adapt to the evolving dynamics of the search tree, while Stage2 provides the flexibility to adjust rules according to tree states.

We also evaluate the impact of asynchronous LLM querying on our framework. Specifically, we implement a sequential querying approach (denoted as Sequ.), where the solver waits for the LLM's response before executing each branching decision whenever a tree state update occurs. As shown in Figure 4(a), this serialized interaction introduces substantial delays, severely degrading solver efficiency compared to all other methods. In contrast, the asynchronous design avoids blocking the solver's progress, effectively masking the latency of LLM inference and enabling smoother integration of decision feedback into the branch-and-bound process. We

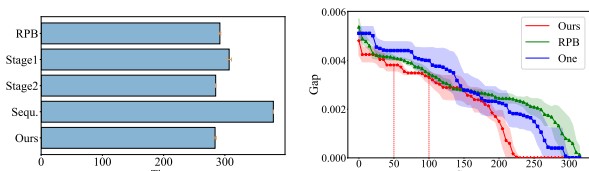

(a) Time-consuming comparison (b) CVisualization of the optimization procedure

Figure 4: Experimental results on the impact of the asynchronous multi-LLM decision mechanism (evaluated on the capacitated facility location benchmark).

also evaluate the single-LLM setting to validate the effectiveness of using multiple LLMs. In this setup, we adopt Qwen-30B as the sole model due to its strong mathematical reasoning ability. However, as shown in Figure 4(b), the single-LLM approach does not perform well. This is mainly because a single model is more prone to biased judgments, and unstable recommendations under complex and evolving tree states. In contrast, the ensemble of multiple LLMs, combined with a voting mechanism, mitigates these issues by reducing variance, filtering out unreliable suggestions, and producing more robust branching rule schedules.

## 6 CONCLUSION

This work introduces a dynamic branching rule scheduler that adapts branching strategies to the evolving state of the branch-and-bound tree. The main challenge is to assess the intrinsic suitability of different heuristic branching rules for a given problem and to determine whether the current rule remains effective as the search progresses. To address this, we introduce large language models (LLMs) to evaluate both the applicability of branching rules to specific problems and their effectiveness during the tree search. We further design two specialized prompts to guide the LLM in deciding when and how to switch rules. Unlike approaches that train branching policies from scratch, our method exploits the broad prior knowledge encoded in LLMs, removing the need for costly data collection and training while enabling generalization to unseen problem types. Experiments on four classical MILP benchmarks show competitive performance, underscoring the potential of LLMs to advance MILP solving.

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

**Clarification**: We use LLMs both for exploring MILP solver optimization and for refining the writing. Additional details are provided in the paper.

# A PRELIMINARY EXPERIMENTS

Figure 5 presents results on three additional MILP problems, each solved with five different branching rules provided by SCIP. As shown, the solving times vary substantially across rules, underscoring the sensitivity of solver performance to the choice of the initial branching rule.

Figure 6 illustrates the impact of rule switching across different problem types. For instance, in the Maximum Independent Set problem (Figures 6(a), (d)) and the Set Covering problem (Figures 6(b), (e)), switching branching rules during the B&B process leads to achieving optimal solutions with fewer branching steps. Moreover, Figures 6(c), (f) show that both the choice of the initial rule and the selection of the rule to switch to are crucial for optimization. These observations motivate our focus on rule selection throughout the B&B process.

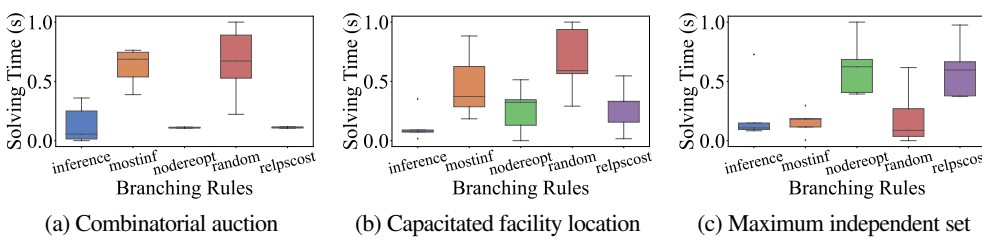

|(a) Combinatorial auction | (b) Capacitated facility location | (c) Maximum independent set |

Figure 5: Time-consuming comparison among different branching rules on three datasets.

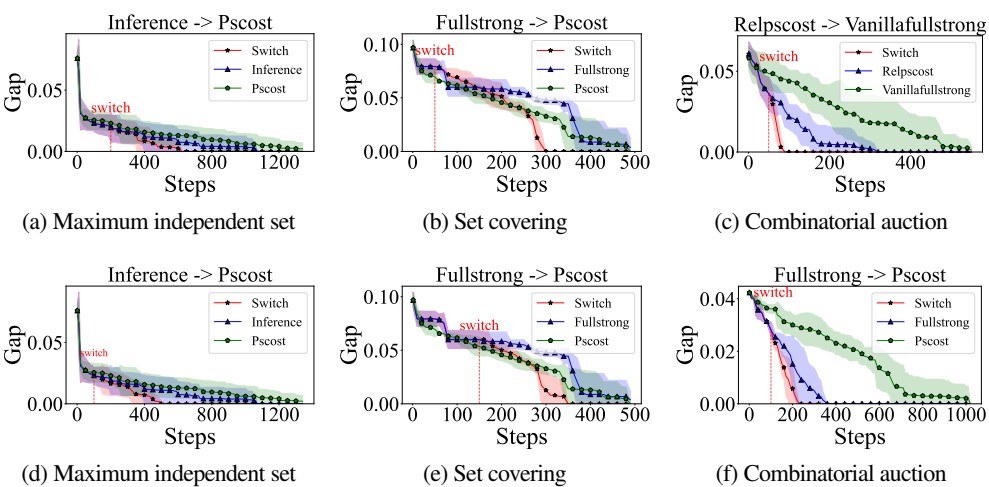

(a) Maximum independent set     (b) Set covering     (c) Combinatorial auction

(d) Maximum independent set     (e) Set covering     (f) Combinatorial auction

Figure 6: Branching rule switching performance on different datasets.

# B BRANCHING RULE PROMPT

Figure 7 illustrates the construction of the initial branching rule prompt, which consists of the problem type, available rule descriptions, and problem descriptors. The problem descriptors summarize key structural statistics of the instance, including the total number of variables, the counts of binary, integer, and continuous variables, the total number of constraints, the average number of nonzeros per constraint, whether the model contains indicator constraints, whether the objective function is quadratic, and whether the instance is a mixed-integer program. Figure 8 illustrates the construction of the dynamic rule switching prompt, which consists of three main components: (i) the text-based representation of the current subtree, encoding depth,

---

**Prompt for initial branching rule**

Yeou are an expert in mixed-integer optimization and SCIP solver configuration. Your task is to choose the best branching rule from a known list, based on the following:

    1. Problem type (e.g., set covering, capacitated facility location, combinatorial auctions, maximum independent set).
    2. Problem structural statistics (e.g., number of variables/constraints, LP density, cut usage).
    3. Variable and constraint types (binary-heavy, general integers, continuous coupling, etc.).

Available branching rules and their recommended use cases:

**Problem_type** = {Set covering}

**Available_rules** = {Rule1, Rule2, ... , RuleM}

(1) **Rule1**: Best when: medium-to-deep trees, many nodes already explored so pseudo-costs are informative...

(2) **Rule2**: Most accurate but very expensive; Best when: root or very shallow depth (D≤2~3)...

        **....**

(M) **RuleM**: Best when: problem shows clear variable clustering/community structure...

**Problem info (JSON)**:
```
{
    'number of variables': 500,
    'number of constraints: 1000,
            ....
}
```
 - Select the best matching branching rule to solve the problem efficiently, considering type, structure, and scale.
 - rule_name must exactly match one from the available rule names.
 - Do NOT output explanations, alternatives, or additional fields.
 - Please strictly output ONLY the bare JSON object in this format: {"branching_rule": "rule_name"}

Figure 7: The illustration of initial branching rule prompt.

---

**Prompt for dynamic rule switching**

Yeou are an expert in mixed-integer optimization and SCIP solver configuration. you are given the search history with the following metrics:

Gap[k]: current optimality gap, Nodes[k]: number of explored nodes, Cut[k]: pruning (cutoff) ratio, Dom[k]: domain reduction ratio, Cands[k]: number of branching candidates, H[k]: candidate entropy, Depth[k]: depth of the current node.

Decide whether to SWITCH the branching rule NOW based on BOTH the historical effects above and the current tree state:

| Compute step-wise according to the definition: $\Delta$Gap = Gap[k] - Gap[k-1]; $\Delta$Nodes = Nodes[k] - Nodes[k-1]  Per-node improvement r = $\Delta$Gap / max(1, $\Delta$Nodes) | Tree-to-text representation: 1 \| Depth: 0 \| Nodes: 1 \| Gap: 0.25 \| Cut: 0.00 \| Dom: 0.02 \| Cands:10 \| H: 2.20  2 \| Depth: 1 \| Nodes: 2 \| Gap: 0.20 \| Cut: 0.10 \| Dom: 0.04 \| Cands: 8 \| H: 1.95  3 \| Depth: 2 \| Nodes: 3 \| Gap: 0.17 \| Cut: 0.15 \| Dom: 0.06 \| Cands: 6 \| H: 1.60   ... ... |

**Decision rubric:**

**1. Keep current rule if progress is acceptable:**

  - $\Delta$Gap ≤ -1e-3 (gap is shrinking), or

  - r ≤ -1e-5 (improvement per node is positive), or

  - Cut[k] is increasing compared to the previous window.

**2. Consider switching rule if stagnation occurs:**

  - If $\Delta$Gap ≥ -1e-3 AND r ≥ -1e-5 (gap not improving), AND

  - Progress metrics (Cut, Dom) are worse than in the previous window.

**3. Rule switching heuristics (choose one from candidates):**

  - If Depth ≤ 3 and Cands ≥ 50 → prefer `fullstrong` or `lookahead`.

  - If Dom is high but Cut is low → propagation not converting to pruning → choose `relpscost`.

  - If Dom and Cut are both low → diversify with `mostinf` or `distribution`.

  - If Depth ≥ 15 (deep tree) → avoid expensive rules, choose `relpscost` or `pscost`.

  - If entropy H is very high → choose `distribution` or `multinode`.

 - Select the best matching branching rule to solve the problem efficiently, considering the tree state.
 - rule_name must exactly match one from the available rule names.
 - Do NOT output explanations, alternatives, or additional fields.
 - Please strictly output ONLY the bare JSON object in this format: {"branching_rule": "rule_name"}

Figure 8: The illustration of dynamic rule switching prompt.

---

explored nodes, gap, cutoff ratio, domain reductions, candidate size, and entropy; (ii) decision rubrics that compare sliding-window trends of these metrics to assess whether the current branching rule remains effective and heuristic guidelines that map different tree states to suitable candidate rules. Together, these elements instruct the LLM to reason about rule effectiveness and determine whether a switch is necessary, and if so, which rule should be activated for subsequent exploration.

