# OpenReview forum: "Large Language Model Guided Dynamic Branching Rule Scheduling in Branch-and-Bound"
_ICLR.cc/2026/Conference — Submitted to ICLR 2026_

### Official Review · Reviewer_52XC · 2025-10-29

**Soundness:** 3
**Presentation:** 2
**Contribution:** 3
**Rating:** 4
**Confidence:** 4

**Summary:**

This paper proposes an innovative framework that integrates Large Language Models (LLMs) into the Branch and Bound (B&B) algorithm to improve decision-making in combinatorial optimization. Specifically, the authors employ LLMs to assist in node selection, branching, and pruning decisions, aiming to reduce search space and accelerate convergence. The idea of incorporating high-level reasoning into a classical exact optimization algorithm is conceptually intriguing and demonstrates a novel cross-disciplinary perspective between symbolic optimization and neural reasoning.

However, while the conceptual direction is interesting, the practical feasibility remains highly questionable. The major concern lies in the extremely high computational overhead of invoking an LLM at every decision step within B&B. Given that B&B may expand thousands or even millions of nodes, the time and resource consumption quickly become prohibitive. The paper currently lacks a discussion or analysis on how to mitigate this issue, such as through model distillation, caching, or selective LLM querying. As a result, the proposed framework appears difficult to scale beyond small toy instances.

**Strengths:**

1. Creative integration of LLMs and classical optimization: The work explores a fresh and potentially impactful direction that bridges symbolic search and neural reasoning.

2. Clear methodological presentation: The paper describes how the LLM fits into the B&B pipeline in a systematic way.

3. Empirical feasibility on benchmarks: Initial experiments show that LLM-guided decisions can lead to improved pruning and shorter search depth.

**Weaknesses:**

1. Severe computational overhead: Calling an LLM (especially large models like GPT-4) at every B&B step is computationally infeasible for realistic problem sizes.

2. Lack of efficiency analysis: The paper does not quantify runtime costs or provide complexity estimates of LLM usage.

3. Limited scalability: Experiments are only conducted on small-scale problems; the method’s practicality for larger MILP instances is unverified.

4. No mitigation strategy: The paper lacks any discussion of reducing LLM inference cost (e.g., distillation, caching, or hybrid heuristics).

**Questions:**

See Weaknesses

---

### Official Review · Reviewer_uKBr · 2025-10-31

**Soundness:** 2
**Presentation:** 4
**Contribution:** 3
**Rating:** 4
**Confidence:** 3

**Summary:**

The paper notes that a single branching rule used throughout B&B is often sub-optimal because the tree structure evolves. The authors propose to let large language models dynamically schedule rules:
1) At the root, an ensemble of LLMs votes for an initial rule based on problem type and size;
2) During search, every L steps the recent sub-tree is converted into text and the LLMs decide whether to switch rules;
3) Asynchronous queries and majority voting reduce latency and hallucination.
On four NP-hard benchmarks the method beats SCIP’s default RPB and four learning-based branching policies in solving time, without any training.

**Strengths:**

1. Training-free generalisation to unseen problem types, avoiding the data-hungry nature of ML-based branching.
2. Practical asynchronous + ensemble mechanism improves robustness and hides LLM latency.
3. Consistent speed-ups over SCIP default and recent learning baselines on four representative problem classes.

**Weaknesses:**

1. Prompts require manual curation of extensive rule descriptions; maintainability and extensibility are not discussed.
2. Only schedules existing SCIP rules; coupling with other commercial solvers is not studied.
3. No theoretical guarantees, e.g., regret bounds or convergence analysis of the scheduling policy.

**Questions:**

1. If all LLMs hallucinate the same poor rule, is there a fallback safeguard?
2. Asynchronous advice may arrive tens of nodes late—does this still guide the search effectively, and could an adaptive trigger frequency help?
3. Have you tried smaller open-source models (e.g., 7B) to reduce cost, and how much performance is lost?

---

### Official Review · Reviewer_j5DD · 2025-11-01

**Soundness:** 2
**Presentation:** 3
**Contribution:** 2
**Rating:** 4
**Confidence:** 4

**Summary:**

The paper proposes an LLM-guided dynamic branching rule scheduler for branch-and-bound in MILP. It selects an initial branching rule from problem descriptors and adaptively switches rules during search via tree-to-text prompts, asynchronous multi-LLM querying, and voting. Experiments on SC/CA/CFL/IS compare against SCIP’s reliability pseudocost (RPB) and ML baselines (SVMRANK, LMART, GCN, tMDP).

**Strengths:**

- Addresses an important and underexplored question: rule scheduling conditioned on evolving tree state.
- Practical design: tree-to-text representation, asynchronous queries, multi-LLM voting.
- Competitive results on multiple benchmarks without training data.

**Weaknesses:**

- Fairness of comparisons is questionable. GCN/tMDP are constrained to CPU inference, while the proposed method appears to rely on external LLM APIs (effectively offloading compute). If the baselines’ GPU is disabled, the LLMs should also be forced to local CPU inference or their API latency/compute must be counted explicitly. Otherwise the test-time “zero-training” advantage is conflated with outsourced compute.
- Missing comparisons to prior LLM-for-BnB work. There is a growing body of agentic/LLM methods for MILP/BnB (e.g., LLM4Solver and related), and the paper does not include head-to-head results, weakening credibility of the claimed benefits of LLM-guided scheduling.
- Limited ablations on scheduling frequency/cost. The number of LLM calls, end-to-end latency impact, and sensitivity to prompt design are not quantified.

[1]  LLM4Solver: Large Language Model for Efficient Algorithm Design of Combinatorial Optimization Solver

**Questions:**

- Will you enforce a fair compute protocol? For example: (i) deploy the LLM ensemble locally on CPU (or a fixed on-prem GPU) and include its inference time in the reported wall-clock; or (ii) if using API, report per-instance number of calls, p50/p95 latency, total API time, and treat it as part of solving time. Alternatively, allow GCN/tMDP to use GPU so all methods leverage external accelerators.
- Can you add direct comparisons to existing LLM-based BnB methods (e.g., LLM4Solver and related agentic solvers) under the same datasets and limits?
P- lease report the scheduling overhead: average calls per instance, decision adoption rate, and how solving time changes if switching is disabled or made sequential (blocking) across all datasets.

---

### Meta-Review · Area_Chair_TYty · 2025-12-15

**Summary:**

The reviewers are mainly concerned about, 1) the fairness setting of CPU/GPU inference, missed LLM baseline; 2) possibility of integration with commercial solvers; 3) heavy computational overhead, lack of efficiency analysis and limited scalability.  Leaving the reviewer comments behind, I like the topic and idea of this work. Unfortunately and surprisingly, the authors did not rebuttal, and this paper thus cannot be accepted given the outstanding concerns (though they are addressable).

**Reviewer Concerns:**

The authors did not rebuttal at all, so I assume all the above mentioned concerns would be remained.

**Reviewer Scores:**

The original scores are 4, 4, 4. Since the authors did not rebuttal, the scores would keep unchanged.

---

### Decision · Program_Chairs · 2026-01-26

Reject